Hardly habitual: chimpanzees and gorillas show flexibility in their motor responses when presented with a causally-clear task

Jacobson Sarah L. 1 2
Hopper Lydia M. lhopper@lpzoo.org 2
1 Psychology, City University of New York, Graduate School and University Center , New York , NY , United States of America
2 Lester E. Fisher Center for the Study and Conservation of Apes, Lincoln Park Zoo , Chicago , IL , United States of America
Vonk Jennifer
Electronic publication date: 2019 Jan 8
Publication date: 2019
Volume: 7
Electronic Location ID: e6195
Received 2018 Sep 21; Accepted 2018 Dec 1
Copyright: ©2019 Jacobson and Hopper
Copyright year: 2019
Copyright holder: Jacobson and Hopper
License: This is an open access article distributed under the terms of the Creative Commons Attribution License, which permits unrestricted use, distribution, reproduction and adaptation in any medium and for any purpose provided that it is properly attributed. For attribution, the original author(s), title, publication source (PeerJ) and either DOI or URL of the article must be cited.
License URL: https://creativecommons.org/licenses/by/4.0/

Keywords: Flexibility, Conservatism, Causal understanding, Habit formation, Gorilla, Chimpanzee

Funding: Leo S. Guthman Fund This work was supported by the Leo S. Guthman Fund. The funders had no role in study design, data collection and analysis, decision to publish, or preparation of the manuscript.

==============================
In contrast to reports of wild primates, studies of captive primates’ flexibility often reveal conservatism: individuals are unable to switch to new and more efficient strategies when task demands change. We propose that such conservatism might be a result of task design and hypothesize that conservatism might be linked to primates’ lack of causal understanding in relation to experimental apparatuses. We investigated if chimpanzees (Pan troglodytes) and western lowland gorillas (Gorilla gorilla gorilla) would show greater flexibility when presented with a causally-clear task. We presented six chimpanzees and seven gorillas with a clear tube from which they had to remove straws to release a reward. To first evaluate the apes’ causal understanding, we recorded the efficiency with which the apes solved the task (i.e., whether they only removed straws below the reward, ignoring redundant ones above it). To further explore how they solved the task, we also recorded the order in which they removed the straws, which allowed us to determine if habitual action sequences emerged. All apes spontaneously solved the task in their first trial and across repeated trials the majority of their solutions were efficient (median = 90.9%), demonstrating their understanding of the puzzle. There was individual variation in the consistency of straw removal patterns exhibited by the apes, but no ape developed an exclusive habit in the order with which they removed the straws, further indicating their causal understanding of the task. Next, we presented the apes with a new configuration of the same task that required the apes to remove fewer straws to obtain the reward. All apes switched to a more efficient straw removal sequence even though their previously-successful, but now less-efficient, solution remained available. We theorize that because the apes understood the causality of the task, they did not form habits and were not conservative.

Introduction

Many primate species are highly flexible. For example, dietary flexibility has been observed in numerous primate species, demonstrating their ability to adopt novel food sources and niches in the wild (e.g.,  Chapman, 1987), although rank and sex may impact their likelihood to do so (Reader & Laland, 2001). Experimental studies have also revealed that chimpanzees can flexibly switch foraging locations to optimize the value of rewards they obtain (Van Leeuwen et al., 2013; Hopper et al., 2015). Beyond what and where to eat, chimpanzees and other primates demonstrate flexibility in how they eat (i.e., foraging techniques, Sanz & Morgan, 2011; Luncz & Boesch, 2014). For example, Sakura & Matsuzawa (1991) observed chimpanzees in Bossou, Guinea, flexibly substituting different materials to use as anvils when cracking palm nuts. Wild capuchin monkeys have also been observed switching the type (size and weight) of stone tools they use to crack open cashew nuts in response to changes in the ripeness (i.e., hardness) of the nuts (Luncz et al., 2016). Similarly, through experimental work, Yamamoto, Humle & Tanaka (2013) demonstrated that captive chimpanzees could switch to a more efficient foraging technique when using straws to get a juice reward after observing a social partner use a more efficient method.

There is, however, mounting evidence of chimpanzees’ lack of flexibility when presented with novel problem-solving tasks in many experimental contexts, both in captivity and the wild (e.g., Hrubesch, Preuschoft & Van Schaik, 2009; Gruber et al., 2011; Manrique & Call, 2015; Harrison & Whiten, 2018), which some have argued explains their reduced complexity of material culture in comparison to humans’ (e.g.,  Dean et al., 2014). Conservatism, the disinclination to adopt an alternative behavior after already learning a successful technique (Hrubesch, Preuschoft & Van Schaik, 2009), although seemingly detrimental, can sometimes be beneficial as it protects an individual from the risk of failure or danger. However, an individual that perseveres in a behavior that is inefficient or suboptimally rewarding, rather than trying other strategies, may not be able to adapt to environmental changes (Brosnan & Hopper, 2014). An early example of chimpanzee conservatism was described by Hrubesch, Preuschoft & Van Schaik (2009) who blocked chimpanzees’ ability to use a shaking solution to retrieve food from a maze device. When this shaking method was prohibited, Hrubesch, Preuschoft & Van Schaik (2009) reported the inability of some of the subjects to use an alternative solution, which they could observe from other conspecifics. Similarly, Manrique, Völter & Call (2013) coaxed apes to adopt novel techniques to solve a puzzle device by changing the configuration of the task and making previously-profitable techniques unsuccessful. However, when presented with the new task configuration, over a quarter of the chimpanzees’ responses represented failed attempts as they persisted on using their previously-learned solution. What causes chimpanzees to show conservatism in certain test settings, while showing great flexibility in others, has fueled much debate.

One possible driver of chimpanzees’ conservatism in certain experimental settings might be the design of the tasks presented to them. Given chimpanzees’ increased problem-solving success in tasks that provide causally-relevant cues, compared to those that provide only arbitrary cues (Hanus & Call, 2011), we theorize that their ability to adopt more efficient strategies may be inhibited when they receive limited or no causal information about a task or its potential solutions (c.f. “ill-structured tasks”, Chappell et al., 2013). Perhaps when chimpanzees cannot discern the causal influences of their actions, they perpetuate them, even when task demands change. Although not a test of individual chimpanzee problem solving and conservatism, results from a study of chimpanzee social learning shed light on the potential interplay between flexibility and causal understanding. Horner & Whiten (2005) showed young chimpanzees how to retrieve a reward from a novel puzzle task using both actions that were relevant and irrelevant to this goal. Importantly, the chimpanzees were either given transparent or opaque versions of the same task—in the former it was clear which actions were required to obtain the reward while in the latter it was not. Perhaps unsurprisingly, those chimpanzees that were given the transparent task copied only the model’s actions that were relevant to obtaining the reward, while those chimpanzees that were given the opaque task “blindly” copied all demonstrated actions, irrelevant or otherwise (see also Whiten et al., 2009). Those chimpanzees that had been given an opaque task initially were later given a transparent version of the task. Instead of perpetuating the action sequence that they had initially adopted to obtain the reward, and which included both relevant and irrelevant actions, when the chimpanzees now received causal information with the provision of the transparent task, they switched to a new action sequence, and one that was more efficient.

We propose that presenting apes with a causally-clear task should protect them from conservatism. To test this, we designed a novel task that was transparent, so that subjects could observe the outcomes of their interactions with the task (sensu Yamamoto, Humle & Tanaka, 2013; Davis et al., 2016), and one that we predicted would be causally clear to the subjects, taking advantage of apes’ understanding of gravity (e.g.,  Tomonaga et al., 2007). The task was a transparent tube with paper straws threaded through it that prevented a food reward from falling down the tube. To access the reward, subjects had to pull out the straws below the reward. The number of straws below (relevant) and above (irrelevant) the reward could be changed across test phases (Fig. 1). While many previous tasks used to investigate flexibility have required changes in distinct types of solutions (e.g., Lehner, Burkart & Van Schaik, 2011; Manrique, Völter & Call, 2013), we took a novel approach and looked to see if apes could change the number of actions required to solve a task. Although this is a less distinct change, and therefore conservatism may seem less likely, we note the findings of previous research that have reported chimpanzees’ failure to change the type of token they exchanged in order to get a better reward even though the action itself (token exchanging) remained constant and so was familiar (e.g., Hopper et al., 2011; Vale et al., 2017).

Figure 1 The Ape-plunk apparatus.

(A) Schematic of the Ape-plunk apparatus (not to scale) in the two different configurations: phase 1 configuration with four straws below the reward shown on the left, phase 2 configuration with two straws below the reward shown on the right. Each numbered cylinder represents a paper straw and the black circle represents the food reward. The apes only needed to remove the straws below the food reward to cause it to fall so they could obtain it, but it was possible for the apes to pull out any and all of the five straws in both phases. (B) One of the gorilla subjects with the apparatus in phase 1 viewed from the experimenter’s area. The apparatus was attached to the outside of the mesh and the bottom of the tube fed into their enclosure so that any rewards that fell out of the apparatus, fell into the apes’ enclosure.

To gain a more comprehensive perspective on ape conservatism, rather than simply testing a single species, as many previous tests in this field have done (e.g., Hrubesch, Preuschoft & Van Schaik, 2009; Lehner, Burkart & Van Schaik, 2011), we tested both chimpanzees (Pan troglodytes) and western lowland gorillas (Gorilla gorilla gorilla) following the same protocol. Gorillas’ tendency for conservatism has not been investigated as extensively as for chimpanzees; two studies have tested three gorillas in different tasks, both revealing some inflexibility when task conditions changed and some success in overcoming a change in conditions (Manrique, Völter & Call, 2013; Manrique & Call, 2015). The differences between chimpanzees’ and gorillas’ performance were insignificant in these studies. We wanted to continue this species comparison with a larger sample of western lowland gorillas and directly compare to chimpanzees housed in the same zoo by following the same protocol (sensu Griffin & Diquelou, 2015).

Our prediction was that if apes had causal understanding of a novel task they would not show conservatism. Therefore, to validate that our novel task was causally clear, our first aim was to test the apes’ causal understanding of the task. To do so, we tested whether the apes could spontaneously efficiently retrieve the reward when provided with the task in the first configuration (four straws below the reward, one above it) by removing only the straws below the reward. As a second indicator of causal understanding, we also documented the order of straw removal the apes used in each trial to determine if the apes formed action habits when solving the task in the first configuration. Specifically, we propose that habit formation might be indicative of reduced causal understanding because habitual actions are based on antecedent stimuli rather than the causal consequences of each action (Dezfouli & Balleine, 2013). We also predicted that repeatedly using the same “action sequence” might be associated with reduced flexibility. Our second aim was to evaluate the apes’ flexibility. To do so, we tested the apes’ responses after they had demonstrated consistent efficiency in solving the task, by changing the configuration of the task (two straws were below the reward, and three above it). In this second configuration a more efficient solution was possible (remove only two straws), while the original, but now less-efficient, strategy (remove four straws) remained available. We did not predict a difference in flexibility between chimpanzee and gorillas since the minimal previous investigations with both species have not seen differences in performance (Manrique, Völter & Call, 2013; Manrique & Call, 2015).

Materials & Methods

Subjects and housing

The subjects were six chimpanzees (five females, age range 17–34, average 25 years) and seven western lowland gorillas (three females, age range 4–27, average 16 years) (Table 1). All of the apes were socially-housed in groups living in naturalistic exhibits at Lincoln Park Zoo, Chicago, USA. All the exhibits in which the apes were housed included an outdoor yard and an indoor dayroom that featured a variety of climbing elements (trees, bamboo, and vines) and deep-mulch substrate.

Table 1 The apes at Lincoln Park Zoo who participated in the study.

Ape name (sex, age)	Species	Social group	Tested in social setting?	
CA (F, 32)	Pan troglodytes	C	Y	
CH (F, 17)	Pan troglodytes	C	Y	
CO (F, 34)	Pan troglodytes	D	Y	
KY (F, 26)	Pan troglodytes	C	Y	
MA (F, 26)	Pan troglodytes	D	Y	
OP (M, 17)	Pan troglodytes	C	Y	
AM (M, 11)	Gorilla gorilla gorilla	B	N	
AZ (M, 13)	Gorilla gorilla gorilla	B	Y	
BA (F, 22)	Gorilla gorilla gorilla	A	Y	
BH (F, 26)	Gorilla gorilla gorilla	A	Y	
KW (M, 27)	Gorilla gorilla gorilla	A	Y	
MO (M, 10)	Gorilla gorilla gorilla	B	N	
PA (F, 4)	Gorilla gorilla gorilla	A	Y	

Ethical note

This study was approved by the Lincoln Park Zoo Research Committee, which is the governing body for research at Lincoln Park Zoo. This research adhered to legal requirements in the United States of America and to the ASAB/ABS Guidelines for the Use of Animals in Research. The apes were tested in their social group on exhibit, except for two gorillas who were tested individually when voluntarily separated as part of daily husbandry routines. All testing was voluntary for the subjects and no changes were made to their typical husbandry routine for this study. The subjects were never food or water deprived. The food rewards and straws used for the experimental protocol were approved by veterinary staff.

Apparatus

The “Ape-plunk” apparatus (named with a nod to the children’s game KerPlunk, Mattel Inc.) was a clear PVC tube (56 cm long, 3.2 cm in diameter) attached to the outside of the cage mesh (Fig. 1). Five holes, spaced 10.2 cm apart, were drilled perpendicular to the tube, such that white, dye-free paper straws (Sweets and Treats©) could extend through the tube and cage mesh so that they were accessible to the ape. Paper straws varied in length between gorillas and chimpanzees due to safety considerations: for test sessions with gorillas the straws were 19.7 cm long and for sessions with chimpanzees the straws were 38 cm long.

Procedure

To begin a trial, we inserted paper straws through the tube from the bottom to the top, but not far enough so the ape could access them, and baited the Ape-plunk with a high-value food reward (either a grape or peanut in-shell depending on the subject’s preference), such that it rested on one of the straws threaded through the tube (Fig. 1). Once all straws were inserted, we used a stiff flat board to push all the straws forward simultaneously until all protruded 3–5 cm into the apes’ enclosure (this was done to avoid any cueing that might have encouraged the subjects to select specific straws). A trial started when all straws were pushed into reach of the subject. The subject could then pull out any straws above or below the reward. A trial ended when the reward fell into the subject’s exhibit and we then removed any remaining straws.

We ran two sequential phases with each subject. For each trial in phase 1, we set up the Ape-plunk with four straws below the reward, and one above it (configuration 1, Fig. 1). Following previous studies of ape flexibility (e.g., Davis et al., 2016), phase 1 continued until the subject was consistently efficient, completing at least 20 trials over multiple sessions (except for one subject, PA, who completed only 17 efficient trials due to experimenter error). Consistent efficiency was defined as the four straws removed below the reward to successfully retrieve it (order was not relevant, although was recorded) for greater than 80% of trials (i.e., number of trials was not pre-determined). Once a subject completed phase 1, they were tested in phase 2. In each trial of this phase, we configured the Ape-plunk with two straws below the food reward and three above it (configuration 2, Fig. 1). Each subject had to complete 20 trials in phase 2, regardless of strategy used.

Apes had access to the baited Ape-plunk for no more than 30 minutes per day and subjects received no more than five test sessions per week. For both phases, subjects completed on average three trials per session (SD = 3) and each subject completed phases 1 and 2 in an average of 11 sessions (SD = 4). All testing was completed between June 2016 and September 2017.

To prevent excessive straw consumption, or the apes from using straws as tools to retrieve rewards in later trials, a keeper asked the apes to trade straws back before the next trial began. The subjects were given lower-value food rewards (e.g., pieces of apple) than the reward in the Ape-plunk in exchange for trading the straws to ensure that they did not begin to value straws more than solving the task.

Although many subjects were tested while in their social group to ensure all testing was voluntary, enhancing validity and welfare (sensu Cronin et al., 2018), we “stationed” them and/or provided more than one apparatus to limit social influence (sensu Carter et al., 2012). For individuals tested in a social group, if one ape began a trial (i.e., had removed one or more straws) and another group member began interacting with the Ape-plunk, the trial was aborted and not recorded. A trial was also aborted if the subject walked away without solving the task or if there was experimenter error. On average, 4% of trials per subject were aborted. We also recorded if and when subjects observed their groupmates using the apparatus, to determine the degree to which they received social information. We defined “observers” as those actively attending to the subject if they were within one meter and oriented towards the subject when actively removing straws from the apparatus (sensu Hopper et al., 2007; Hopper et al., 2015). In phase 1, chimpanzee and gorilla subjects each observed two completed trials on average. In phase 2, chimpanzee subjects each observed fewer than one successful trial (0.67 trials) on average, and gorillas each observed just over one successful trial (1.4 trials) on average.

Coding and analysis

The sessions were recorded using a Sony Handycam (HDR-CX160) on a tripod that was set up to record each Ape-plunk available and the participating ape. All data were transcribed into Excel and are available in the supplementary materials. A second coder, blind to the experimental condition, recorded the order of straws removed according to straw position for one randomly-selected session of each phase per subject, and had 100% agreement with the primary coder.

The apes’ responses in a given trial were either classed as efficient (they removed only relevant straws below the reward) or inefficient (they removed both relevant and irrelevant straws) to obtain the reward. In phase 1, there were a total of 120 different possible orders that the apes could remove all five straws, 24 (20%) of which would be efficient, removing the four straws below the reward first. In phase 2, only 12 (10%) of the 120 possible combinations would be efficient, removing the two straws below the reward first.

For all trials, we also recorded the order in which the subject removed the straws using the numbering system in Fig. 1. Each straw-removal order (e.g., 1,2,3,4 or 4,5,3,2,1) was termed an ‘action sequence’. With these data, we determined the diversity of action sequences each individual used in phase 1 and phase 2 by calculating their Shannon’s H diversity index (Shannon & Weaver, 1949; Shepherdson et al., 1993). If subjects repeatedly used the same action sequence (i.e., developed a habit), their diversity index would be lower than those that did not. Due to the small sample sizes in this study, non-parametric statistics were used for all analyses. We used Wilcoxon rank-sum tests to compare the subjects’ action sequence diversity across species and Wilcoxon signed-rank tests to compare diversity across phase. Using the apes’ action sequence pattern data, we also calculated each subject’s median action sequence run length, where a ‘run’ was defined as the subject using the same action sequence in two or more sequential trials. We then used Wilcoxon rank-sum tests to compare the subjects’ run lengths across species and Wilcoxon signed-rank tests to compare across phase.

All analyses were conducted in R version 3.2.3 (R Core Team, 2015). All graphs were plotted with “ggplot2” (Wickham, 2009).

Results

Causal understanding

All 13 apes successfully obtained the reward from the Ape-plunk in their first trial, and 11 (five chimpanzees and six gorillas) did so efficiently (i.e., removing only the four straws below the reward). Across all their trials, the chimpanzees were efficient for a median of 90.9% of their trials and the gorillas were efficient for a median of 95.2% of trials in phase 1 (Fig. 2). There was no difference in efficiency between chimpanzees and gorillas (Wilcoxon rank test: W = 18, N = 13, p = 0.72). Collectively, the apes removed only relevant straws for a median of 90.9% of their trials and four of the 13 apes used an efficient strategy in every trial of phase 1 (Table 2). The apes reached consistency in their efficiency in 22 trials on average (SD = 2); the maximum number of trials needed was 26.

Figure 2 The number of trials in which each action sequence, both efficient and inefficient, was used by the chimpanzees and gorillas in phase 1.

Table 2 Order of straw removal in phase 1 for each of the apes’ trials.a

		Chimpanzees	Gorillas	
		CAb	CH	CO	KY	MA	OP	AMb	AZ	BA	BHb	KW	MOb	PA	
Trial	1	4,3,2,1	4,2,3,1	5,4,3,2,1	1,3,2,4	4,3,2,1	4,3,2,1	1,2,3,4	1,2,3,4	1,3,4,2	2,1,3,4	2,1,3,4	3,4,2,1	3,2,1,5,4	
2	2,1,3,4	4,3,2,1	4,5,1,2,3	4,3,2,1	2,4,3,1	1,4,3,2	1,4,3,2	1,2,4,3	1,4,3,2	3,4,2,1	2,1,3,4	1,2,3,4	
3	1,2,3,4	1,4,3,2	4,3,2,1	1,2,3,4	4,3,2,1	4,3,2,1	2,1,3,4	
4	3,4,2,1	3,4,2,1	5,4,3,2,1	1,2,3,4	1,2,4,5,3	1,2,3,4	3,2,4,1	1,4,3,2	2,3,4,1	1,3,2,4	
5	4,1,3,2	4,3,2,1	4,3,2,1	4,3,2,1	4,2,1,3	2,4,3,5,1	4,3,2,1	2,4,3,1	4,3,2,1	1,2,3,4	4,3,2,1	
6	1,4,3,2	4,3,2,1	4,1,3,2	2,3,1,4	4,3,1,2	1,3,4,2	4,3,2,1	1,2,4,3	1,2,3,4	2,1,3,4	2,3,4,1	
7	4,3,2,1	4,3,2,1	1,5,4,3,2	2,1,3,4	1,4,3,2	1,2,3,4	1,2,3,4	1,4,3,2	5,4,3,2,1	1,2,3,4	2,4,1,3	
8	2,1,3,4	4,1,3,2	4,3,2,1	2,5,4,1,3	1,2,4,3	1,3,4,2	2,1,3,4	2,4,3,1	4,5,3,2,1	4,2,1,3	2,1,3,4	
9	4,3,2,1	2,4,3,1	1,4,3,2	5,4,3,2,1	1,2,4,3	1,2,3,4	3,1,2,4	3,2,1,4	4,3,2,1	3,4,2,1	2,1,3,4	
10	4,3,2,1	4,3,2,1	5,3,4,2,1	1,4,3,2	3,2,1,4	2,1,3,4	1,2,3,4	1,2,3,4	2,1,3,4	4,2,1,3	3,4,1,2	
11	1,2,3,4	4,3,2,1	3,4,2,1	2,1,3,4	2,3,1,4	4,1,2,3	4,3,1,2	4,3,2,1	4,5,2,3,1	
12	2,4,3,1	5,4,3,2,1	1,2,4,3	4,3,1,2	1,2,3,4	1,2,3,4	4,3,2,1	4,3,1,2	
13	5,4,3,2,1	3,4,2,1	2,1,3,4	4,3,2,1	1,2,4,3	2,3,4,1	2,1,3,4	2,3,1,4	4,2,1,3	1,3,4,2	
14	2,3,4,1	4,3,2,1	4,3,2,1	1,2,3,4	1,2,3,4	3,4,2,1	4,2,1,3	2,1,3,4	3,2,1,4	2,1,3,4	2,3,4,1	
15	4,3,2,1	4,5,3,1,2	4,1,2,3	2,3,4,1	1,2,3,4	4,3,2,1	3,1,4,5,2	2,4,3,1	4,3,1,2	
16	1,2,4,3	4,3,2,1	1,2,4,3	1,2,3,4,5	1,2,3,4	4,3,2,1	3,4,2,1	4,2,3,1	
17	4,3,2,1	4,3,1,2	4,3,1,2	4,3,2,1	2,1,3,4	4,2,3,1	5,4,1,2,3	1,2,4,3	2,1,3,4	2,3,4,1	
18	4,3,2,1	1,2,3,4	4,1,3,2	1,2,3,4	1,2,3,4	4,5,3,2,1	1,2,3,4	4,3,2,1	
19	4,3,2,1	4,3,2,1	1,3,4,2	5,4,3,2,1	2,3,4,1	
20	4,5,3,2,1	1,2,3,4	2,3,4,1	1,2,3,4	1,2,3,5,4	2,1,3,4	3,4,2,1	4,3,2,1	3,4,5,2,1	
21		4,3,2,1	4,3,1,2	1,2,3,4		3,4,2,1	1,2,3,5,4			
22			4,2,3,1	3,4,2,1			1,2,3,4		4,3,2,1			
23				4,3,2,1			3,1,2,4				
24										3,2,4,1			
25										3,1,4,2			
26										4,2,3,1			
H index	1.37	1.11	1.45	2.23	1.37	1.83	1.96	1.70	1.95	1.75	2.36	1.77	2.32	
Median run length	4	3	2	2	4	5	2.5	3	2	4	2	2	2	
Max run length	5	6	6	2	7	8	4	4	3	4	2	3	2	
Notes.

a Straws were numbered from bottom to top, as illustrated in Fig. 1. Runs (>1 trial in a row with the same action pattern) are merged and highlighted in grey.

b Four apes (CA, AM, BH, and MO) used an efficient action sequence in every trial of phase 1.

Action sequences in phase 1

The action sequences (i.e., order of straw removal) the apes used when they solved the task in phase 1 varied widely between and within individuals (Table 2, see example footage here of a chimpanzee using three different action sequences to solve the task in phase 1: https://figshare.com/articles/Chimpanzee_problem_solving_and_flexibility/6654896/2). Collectively, the apes used 38 different action sequences to solve the task (120 were possible) (Fig. 2). Specifically, the apes used 21 of the 24 possible action sequences that were efficient, removing straws 1, 2, 3, and 4 first (e.g., 1,3,2,4 or 4,3,2,1), but only 12 of the 96 possible action sequences that were inefficient in which they pulled out the irrelevant straw before releasing the reward (e.g., 5,4,3,2,1 or 3,5,4,2,1). The chimpanzees used a total of 16 different efficient action sequences and nine different inefficient action sequences, while the gorillas used 22 different efficient action sequences and nine different inefficient action sequences. The apes also continued to try new action sequences throughout phase 1, across trials (Fig. 3).

Figure 3 The cumulative number of different action sequences used by each chimpanzee (A) and gorilla (B) in phase 1.

Not only was there variance across individuals as to the way in which they solved the task, there was also variance in the number of different action sequences each individual used across trials, ranging from seven to 14. Indeed, the apes’ individual Shannon H index ranged from 1.11 to 2.36 (Table 2), where the index would be 0 if only one action sequence was used and 3.63 if each possible sequence was used once. There was no significant difference between the chimpanzees’ H index scores (median = 1.41) and the gorillas’ (median = 1.95; Wilcoxon rank-sum test, W = 8, N = 13, p = 0.07). The apes’ median run length ranged from 2–5 trials (Table 2) and did not differ by species (Wilcoxon rank-sum test, W = 29.5, N = 13, p = 0.23).

The action sequence most commonly used by the chimpanzees in phase 1 was repeatedly removing the straw directly below the reward (i.e., 4,3,2,1), which occurred for a median of 60.8% of individuals’ trials (Fig. 2). In contrast, this action sequence was used by gorillas for a median of 9.5% of their trials. The proportion of trials in which chimpanzees used the 4,3,2,1 response pattern was significantly different from the gorillas (Wilcoxon rank-sum test: W = 169, N = 13, p < 0.01). In contrast to chimpanzees, the gorillas most frequently removed straws from the bottom of the apparatus until they reached the reward (i.e., 1,2,3,4) (median= 35.0%), however they did not use this strategy significantly more than the chimpanzees (median = 4.3% of trials, Wilcoxon rank-sum test: W = 8, N = 13, p = 0.07).

Flexibility

In their first trial of phase 2, when presented with the new configuration of the Ape-plunk, all 13 apes removed fewer than four straws to obtain the reward (i.e., the number that needed to be removed to be efficient in phase 1). Furthermore, 12 of the 13 apes switched to the most efficient strategy on their first trial (i.e., they removed only two straws), which was significantly different than chance (binomial test: p < 0.001). For example footage of a chimpanzee switching from an efficient strategy in phase 1 to a different, but efficient, strategy in phase 2, see this video: https://figshare.com/articles/Chimpanzee_problem_solving_and_flexibility/6654896/2. Five of the 13 apes were efficient in every trial in phase 2 (Table 3). In phase 2, chimpanzees were efficient for a median of 92.4% of trials and gorillas were efficient for a median of 100% of trials, which was not significantly different from phase 1 for either species (chimpanzees, Wilcoxon signed rank test: V = 13, N = 6, p = 0.69; gorillas, V = 9, N = 7, p = 0.79).

Table 3 Order of straw removal in phase 2 for each of the apes’ trials.a

		Chimpanzee	Gorillas	
		CA	CH	CO	KY	MA	OP	AM	AZ	BA	BH	KW	MO	PA	
Trial	1	2,1	4,2,1	2,1	2,1	2,1	2,1	2,1	1,2	1,2	1,2	2,1	1,2	2,1	
2	3,2,1	3,2,1	1,2	2,1	3,2,1	3,2,1	
3	1,2	4,1,3,2	2,1	1,2	2,1	2,1	2,3,1	
4	4,5,3,2,1	4,2,1	3,2,1	2,1	2,3,1	2,1	1,2	2,1	
5	2,1	5,3,1,2	2,1	4,2,1	1,2	1,2	2,3,1	
6	1,2	5,2,1	3,2,1	2,1	2,1	2,1	1,2	2,1	
7	2,1	2,1	1,2	3,1,2	2,1	1,2	2,3,1	2,1	
8	1,2	3,2,1	1,2	2,1	2,1	3,2,1	1,2	
9	2,1	4,2,1	1,3,2	
10	1,2	1,2	2,1	2,1	
11	2,1	2,1	2,1	
12	1,2	1,2	1,2	
13	1,2	1,2	2,1	2,1	
14	2,1	2,1	1,2	2,1	1,2	
15	2,1	1,2	3,1,2	2,1	
16	1,2	1,2	1,2	2,1	
17	2,1	2,1	2,1	2,1	
18	4,2,1	1,2	1,2	2,1	3,2,1	
19	1,2	2,1	2,1	1,2	2,1	1,2	2,1	
20	2,1	1,2	
H index	0.79	1.54	0.20	0.82	0.73	0.82	0.89	0.00	0.20	0.65	1.08	0.42	0.83	
Median run length	3	4.5	10	5.5	4	3	3	20	9.5	2	5	2.5	3	
Max run length	4	7	10	9	6	4	5	20	14	6	5	10	8	
Notes.

a Straws were numbered from bottom to top, as illustrated in Fig. 1. Runs (>1 trial in a row with the same action pattern) are merged and highlighted in grey.

Action sequences in phase 2

Collectively, the apes used 11 different action sequences in phase 2. The chimpanzees used nine different action sequences, including the two possible efficient sequences (i.e., 1,2 and 2,1) and seven that were inefficient. The gorillas used six different action sequences, the two efficient ones and four inefficient ones. The chimpanzees’ modal response was the 2,1 action sequence while the gorillas’ modal response was the 1,2 action sequence (Fig. 4). In phase 2, a median of 67.5% of the chimpanzees’ responses, and 35.0% of the gorillas’ responses was the action sequence in which they sequentially removed the straw directly below the reward (2,1). The use of this response pattern differed between species in phase 2 (Wilcoxon rank-sum test: W = 156, N = 13, p < 0.01), revealing that chimpanzees used it more frequently than gorillas. Gorillas used the alternative efficient action sequence (1,2) significantly more often than chimpanzees (Wilcoxon rank-sum test: W = 3, N = 13, p < 0.01). Indeed, one gorilla (AZ) used the 1,2 action sequence for all of his trials (Table 3).

Figure 4 The number of trials in which each action sequence, both efficient and inefficient, was used by the chimpanzees and gorillas in phase 2.

In Phase 2, chimpanzees had a median Shannon H index of 0.80 and the gorillas’ was 0.65 where the index would be 0 if they used only one sequence and if they used all possible solutions 2.37. As in phase 1, their diversity did not differ by species (Wilcoxon rank-sum test, W = 24, N = 13, p = 0.67). Overall the apes’ diversity indices in phase 2 were significantly lower than in phase 1 (Wilcoxon signed-rank test, V = 90, N = 13, p < 0.01), another indicator of their flexibility and efficiency since there were only two efficient action patterns in phase 2 (Fig. 5). When comparing the diversity of action sequences in phase 1 to the apes’ efficiency in phase 2, there was no correlation between these measures (r = 0.03, p = 0.91). This suggests that there was no relationship between consistency of action sequences used in the first phase and ability to employ efficient action sequences when contingencies changed (flexibility). The apes’ median run length ranged from 3–20 (Table 3) and like in phase 1, did not differ by species (chimpanzee median = 4.25, gorilla median= 4.33, Wilcoxon rank-sum test, W = 25, N = 13, p = 0.61). The apes’ median run length was also not significantly different between phase 1 (median = 2.50) and phase 2 (median = 4.00) (Wilcoxon signed-rank test, V = 16.5, N = 13, p = 0.08).

Figure 5 The diversity of the action sequences (H-index) used by each species in each phase.

Discussion

We tested the interplay between chimpanzees’ and gorillas’ causal understanding, habit formation, and problem-solving flexibility. Confirming that they understood the mechanics of the task, all apes successfully retrieved the reward from the Ape-plunk task on their first trial without training or social demonstration. The apes’ consistently efficient responses and lack of habit formation of action sequences in phase 1 further illustrates their causal understanding of the task mechanism and their goal to obtain the reward using the most efficient method available. Their causal understanding also likely facilitated their flexibility: when the task configuration was changed, the apes did not show conservatism in their response technique. When presented with the task in a new configuration in phase 2, all but one of the apes changed from using one of the previously-efficient 4-straw action sequences and switched to a newly more-efficient 2-straw action sequence on their first trial. Furthermore, the apes showed this flexible adoption of a new response strategy even when their previously-learned, but less efficient, action sequences remained available. This is in contrast to previous tests of ape flexibility that have “scaffolded” subjects’ transitions to new solutions by blocking previously-used ones (e.g., Lehner, Burkart & Van Schaik, 2011; Manrique, Völter & Call, 2013; Davis et al., 2016; Harrison & Whiten, 2018).

Not only did the apes show an ability to change the technique they used across phases (remove four straws versus remove two straws), they also showed exploration within phases in terms of the action sequences they used (i.e., order of straw removal), indicating that they did not form immutable habits. The apes expressed a wide range of variation within and between subjects in the action sequences they used to solve the task, continuing to explore new sequences throughout phase 1. In phase 1, the most efficient manner in which the apes could solve the task was to remove all of the four straws below the reward, but they could do so in any order (i.e., the task did not require a specific sequential response, c.f. Whiten, 1998; Whiten et al., 2006). Even though the required action sequence was arbitrary, and the apes could have simply continued to use the first one that they discovered (it was reinforced immediately), the chimpanzees used a total of 16 different efficient action sequences within phase 1, and the gorillas used 22. This exploration (i.e., lack of habit formation) may be explained by their causal understanding of the task (i.e., they did not develop a “superstitious” response pattern, because they understood the task mechanism).

Although the apes showed exploration of the different action sequences, we also noted that certain individuals persisted in using certain sequences, as highlighted by their low diversity (H index) scores and their long median action sequence run lengths. Even though some individuals had these tendencies, they were still able to transition away from their preferred action sequence when the task configuration changed. This is further supported by the reduction in response diversity (H index score) shown by both species in phase 2 compared to phase 1, when fewer combinations of action sequences could be performed while remaining efficient (i.e., the apes prioritized efficiency over previously-reinforced action sequences).

The two species differed with respect to which action sequence they used most commonly, and these preferences shown by the apes potentially reflect a difference in how they learned to solve the task. Chimpanzees most frequently solved the task by repeatedly removing the straw that the reward rested on within a trial (i.e., 4,3,2,1). We theorize that this was because of two mutually-reinforcing factors. The chimpanzees learned the simple association to pull out whichever straw the reward rested on because doing so caused the reward to fall and move closer to the final goal. In this way, the chimpanzees received proximate reinforcement for each action, even though it was only the final one (removing straw 1) that released the reward and so was reinforced with receipt of food. It is likely that the chimpanzees’ attention was initially drawn to the straw with the grape on it due to a form of stimulus enhancement—the straw became a potent stimulus via association with the food reward touching it. Similarly, in phase 2 the chimpanzees’ most commonly-used action sequence (2,1) followed this same rule, accounting for 69% of their responses. Given this, it is possible that the chimpanzees did not understand the task mechanism holistically, but following this simple rule (remove straw that reward rests on) is both what allowed them to solve the task efficiently and change their response pattern flexibly when the task configuration changed. It is worth noting, however, that 46% of the chimpanzees’ responses in phase 1, and 31% of their responses in phase 2, were via action sequences other than 4,3,2,1 or 2,1 respectively. Thus, they did not follow this rule exclusively but were consistently efficient.

In contrast to the chimpanzees, the gorillas’ most commonly-used action sequence in phase 1 was 1,2,3,4 (i.e., pulling out straws from the bottom of the tube to the top sequentially), which represented 26% of all their responses. They used the 4,3,2,1 action sequence (the chimpanzees’ preferred technique) less frequently—it represented only 13% of their trials. Interestingly, the gorillas’ second most common response was 2,1,3,4 (14% of trials), which might represent a failed attempt at 1,2,3,4 (i.e., they initially pulled out straw 2 instead of 1 with their first action but then reverted to the rest of the 1,2,3,4 sequence). Within the 1,2,3,4 action sequence, only the last action is rewarded and reinforced. No changes are triggered when straws 1, 2, or 3 are removed prior to 4. It is unclear what motivated the gorillas to prefer this action sequence. Perhaps they were motivated to remove the straws associated with where they obtained the reward (i.e., straws closest to the bottom of the device from where the rewards fell). In phase 2, at a species level the majority of the gorillas’ responses followed the pattern of removing straws from the bottom upwards—in 54% of their trials they used the 1,2 action sequence—but at the individual level, each gorilla showed a preference for either the 1,2 or 2,1 sequence. Potentially, the gorillas who preferred removing straws sequentially from the bottom of the apparatus in phase 1 (i.e., 1,2,3,4 action sequence) would not need to understand the task to be flexible when the task configuration changed in the second phase. However, due to the variety of methods used by the gorillas across their trials in phase 1, we conclude that this is not the case. In spite of these individual preferences, the gorillas showed greater behavioral diversity in terms of their action sequences than the chimpanzees and so certainly were not bound to a single technique. Greater understanding about how these apes solved the task, and what caused them to explore multiple action sequences, as well as what explains the species differences we observed, is required. For example, is it possible that the species had a different causal understanding of the task (Visalberghi, 1997), or do their responses reflect differences in self-control (e.g., Judge & Essler, 2013)? We think it is unlikely that these species differences resulted from social learning, as there were very few trials that the subjects observed overall in each phase. Also, there were two groups of chimpanzee subjects and two groups of gorilla subjects tested, so even if social learning occurred in one group, it could not account fully for the species differences observed across multiple groups.

If the apes did not have causal understanding of the task, or had not been motivated to obtain the reward efficiently, we might have expected them to simply remove all five of the straws, regardless of the reward location. Although this strategy is inefficient, the effort to remove extra straws is relatively low. Despite this, in phase 1, the apes used an inefficient 5-straw action sequence in fewer than 10% of their trials and, in phase 2, only one ape removed all of the five straws, and only in one trial. Thus, the apes were not only efficient in phase 1, but showed flexibility in their responses such that they continued to use the most efficient responses as new newly-efficient action sequences became available, discarding previously-adopted action sequences. The apes’ drive to obtain the reward in the most efficient way possible is reminiscent of previous studies of ape problem solving and social learning in which they have been shown to adopt the most efficient strategy, even ignoring socially-demonstrated actions if they did not pertain directly to reward retrieval (e.g., Horner & Whiten, 2005).

Although it was a secondary aim of our study to ascertain if habit formation was an indicator of reduced causal understanding, it is likely that our study design did not provide the apes with a long enough exposure for a habitual action sequence to develop in phase 1. In spite of this, we did see runs of action sequences emerging across trials in the apes’ responses. Furthermore, Davis et al. (2016) observed conservatism by chimpanzees after repeating a multi-stepped solution for only 20 trials, although the task in their study did not have as many action sequences possible as the current study and all trials were completed in a single test session. In the future it would be important to look at the action sequences employed over a longer time period to determine if even more habitual responses developed, and whether this varied by species. It would also be informative to use a task that had irrelevant straws below the reward to look at the limits of the apes’ flexibility and further explore the interplay between conservatism and causal understanding. Another future direction could be to explore apes’ responses when presented with a task that was opaque. Without causal feedback, apes might persist with the first successful action sequence, which would become an engrained habit, and would be difficult to shift away from when contingencies of the task changed.

Conclusions

We predicted that if apes have causal understanding of a novel task they would be flexible when task demands changed. The apes’ overall efficiency solving the novel task demonstrated their causal understanding of it, likely supporting their later flexibility. The apes also were flexible in the sequences in which they removed straws to solve the task, and although there was some individual variation, they did not appear to form habits. In conclusion, we theorize that the apes’ causal understanding of the task (perhaps underscored by simple associative learning rules) protected them against habit formation and also against conservatism, but future work is required to understand the species differences we observed in the pattern of the apes’ responses (i.e., action sequences).

Supplemental Information

Data S1 Raw data of each subject’s straw removal method per trial

Click here for additional data file.

We thank Steve Ross and Katie Cronin for their advice on this manuscript and Lace Lively for reliability coding. We would also like to acknowledge Rebecca Williamson who provided feedback on the apparatus design for an earlier version of this task she and Lydia Hopper developed for use with children. We would also like to thank Maureen Leahy, Jill Moyse, Danielle Fogarty, and the animal care staff at Regenstein Center for African Apes, Lincoln Park Zoo, for their assistance in data collection and support of our research.

Additional Information and Declarations

Competing Interests

Author Contributions

Animal Ethics

Data Availability

Lydia M. Hopper is an Academic Editor for PeerJ.

Sarah L. Jacobson conceived and designed the experiments, performed the experiments, analyzed the data, prepared figures and/or tables, authored or reviewed drafts of the paper, approved the final draft.

Lydia M. Hopper conceived and designed the experiments, performed the experiments, analyzed the data, contributed reagents/materials/analysis tools, prepared figures and/or tables, authored or reviewed drafts of the paper, approved the final draft.

The following information was supplied relating to ethical approvals (i.e., approving body and any reference numbers):

Lincoln Park Zoo’s Research Committee approved this research.

The following information was supplied regarding data availability:

The raw data is available in the Supplementary File (including trials used in analyses and those that were not).

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
