# Peer review of "Hardly habitual: chimpanzees and gorillas show flexibility in their motor responses when presented with a causally-clear task"

_PeerJ, doi:10.7717/peerj.6195_

## Round 0.1 · original submission · Minor Revisions

I am very fortunate that I received two very thoughtful and thorough reviews from expert reviewers. The reviewers agree that the data presented here is interesting and merits publication. We all concur that the paper is well written. Both reviewers have very useful suggestions, however, for improving the clarity and impact of the MS. I suggest you take their comments seriously in preparing a revision. I have a few mostly minor comments of my own:
Avoid using "while" and "since" in non-temporal context (e.g., first line of the abstract) Please check throughout.
Please insert commas after i.e., and e.g.,
Why do you present (in the abstract and on lines 95-100) the study as if chimpanzees were the key subjects and compared to gorillas only as an afterthought? Weren't both species of equal importance to study here? It reads as if you are comparing novel chimpanzee data to pre-existing gorilla data, but it is not clear, and later it is apparent that this is not the case. Although I think there is immense value in comparative approaches, it should be done with clear hypotheses about the comparison species. Is there a reason to expect differences or similarities between the two ape species here? If it's simply exploratory, you can be honest about that as data is lacking on both species, but it seems there is a missed opportunity to use their different ecologies in an informative predictive way.
In the abstract, you contrast perseveration with habitual motor responses, but the distinction is unclear.
Do you need the word relevant on line 122?
It seems that your task (order and number of straws removed) taps into a quite different kind of flexibility than the type that would allow them to change behaviors, targets of behavior (types of food) and locations/contexts etc. In your task, they are not required to alter their actual behavior - they're still removing straws. I wonder if you could comment on how comparable that would be to altering the type of behavior directed at the apparatus altogether (e.g, shaking it versus removing straws)?
Were the sessions coded by a secondary coder selected randomly? Were straws recorded by position, color?
On line 247, move the "only" to before 92.3" on line 248. Similarly on line 392, only is also misplaced.
Perhaps I am missing something, but if the reward falls out as soon as the minimum number of relevant straws have been removed, why would they continue to remove more straws, particularly if they were always biased from the beginning to remove straws below the reward only? Wouldn't it make more sense to have distracter straws that weren't structurally relevant for the reward falling but that appeared below the reward in Phase 2? It would have been especially useful to make the straw directly below the reward causally irrelevant (although I'm not sure how that work practically). I think your design biases you to get the result you obtained indicating flexibility. You will need to address this carefully in the revision.
I don't know if it is true that previously learned action sequences were still available if the action sequence was about removing straws from particular locations rather than just the number of straws removed. It seems like you coded order as an indicator of a habitual pattern but then considered the pattern to be just about the number of straws removed, which is not consistent.

·

Basic reporting

The manuscript is very clearly structured and the logic of the arguments is easy to follow. The language is clear, unambiguous, and professional throughout. I commend the authors for sharing an example video file and for producing very clear and helpful figures. The authors refer to the relevant prior literature appropriately.

Background information:

1. The two introductory paragraphs introducing current evidence on behavioural flexibility and conservatism in chimpanzees from both experimental and field studies are very well written, they are detailed and refer to the relevant literature, but are still concise. A minor point here: Currently these first two paragraphs are somewhat unbalanced given that several examples for behavioural flexibility are given in paragraph 1, but no examples for conservatism in paragraph 2. I suggest to divide the sentence from lines 56-60 into two (cut after “Harrison & Whiten, 2018)”) and to insert a sentence or two between these sentences to provide examples (ideally one from captivity and one from the wild).

2a. I have a more important point regarding the third paragraph (lines 69-88). While I find the authors’ suggestion of causal information and habit formation as possible drivers of chimpanzees’ conservatism highly interesting, I would like to encourage them to expand on their theory a bit more, especially given that the theory seems to be novel: The authors say that chimpanzees’ “ability to adopt novel strategies may be inhibited when they receive limited/no causal information about a task or its potential solutions” (lines 72/73). It is somewhat unclear what “novel strategies” means here. Do the authors mean (as I suspect) the more efficient solutions to an “old” task for which one has already learned a solution (thus describing the switch from an old to a more efficient solution) or do they mean any novel solution to a task (i.e., both more efficient solutions to old tasks for which a solution already exists as well as solutions to tasks that have not been encountered before, see the ill-structured tasks as described in Chappell et al., 2013)? This should be clarified, possibly by changing “novel strategies” to “more efficient strategies”.

2b. Second, the authors speculate that “[p]erhaps when chimpanzees cannot discern the causal influences of their actions, they perpetuate them, even when task demands change.” (lines 74/75). Could the authors provide an example from the literature where they think this could have been the case? If they have provided some examples of chimpanzee conservatism in the introduction (see comment above), they could simply refer to one of the examples here.
Furthermore, does this explanation also hold for studies in which chimpanzees are shown a more efficient strategy by a demonstrator? That is, couldn’t one argue that a demonstration by a conspecific already provides the individual with the necessary causal information to solve the task and that it is something else than a lack of causal information that makes chimpanzees fail to adopt a more efficient strategy? A brief statement on the authors’ stance here would help clarify the scope of the theory.

2c. The authors then introduce the concept of habitual responses; again, while the suggestion of a role of habit formation for chimpanzees’ conservatism is intriguing, more information in this paragraph is needed to clarify the assumed theoretical links between habit formation, causal understanding, and conservatism. First, the authors quote Köhler in saying that “chimpanzees’ failure to solve novel tasks” can be due to “crude stupidities arising from habit” (lines 76/77). What I find a bit confusing here (see comment above) is what exactly is meant by “novel tasks”. As far as I understand Köhler, he was describing cases in which chimpanzees bring to a novel task a solution that has worked well repeatedly in a different context, whereas most current experimental studies on chimpanzees’ conservatism seem to examine these animals’ ability to invent a novel, more efficient solution to an old task. In how far are these cases comparable? Is one more the case of previous experience influencing the innovation of a task solution whereas the other describes the possibility of habits that were formed on a task to perpetuate despite changes in this task’s demands?

2d. More importantly, however, the authors need to make clearer how they see habitual responses, causal understanding, and conservatism relate to each other. Are the formation of habits and the lack of causal information two partially independent factors affecting conservatism (with more causal information reducing the likelihood of habit formation), are they mutually exclusive or orthogonal concepts? While I don’t expect the authors to come up with a fully fledged theory, providing some more information would nevertheless be very helpful, as it is unclear throughout the manuscript how exactly these concepts are assumed to interrelate.

Minor issues:

1. Introduction, lines 84-88: You mention the Horner & Whiten (2005) study as support for your suggestion that habit formation might result in conservatism. I struggle to see where the chimpanzees in this study showed perseveration in solving an opaque box or formed a habit in the first place. The authors report that “When subjects […] transferred from the clear box to the opaque box, although there was a slight increase in the reproduction of irrelevant actions, this was not significant” (p. 170); therefore, in my understanding, chimpanzees did not seem to perseverate on the irrelevant actions. In addition, chimps who were presented with the opaque box first did reproduce more of the irrelevant actions, but dropped most of them once they were presented with the clear box. To me, chimpanzees in this condition did not perseverate on the irrelevant actions either. I suggest that you either elaborate on which result from that study you interpret as habit formation and/or perseveration or that you choose a different study that supports better your suggestion that habit formation might result in conservatism.

2. In the abstract, it is written that it was “investigated if this conservatism is due to the perseveration with a specific solution or if it could be attributed to the development of habitual motor responses”. It does not become clear what is meant with perseveration with a specific solution – how is this different from conservatism? What seems to be missing here is perseveration due to what (possibly a more cognitive explanation?).

3. Very minor grammar point: line 276: remove hyphen in “commonly-used”?

Experimental design

The research is in conformity with the ethical standards in the field; the investigation and analysis was rigorous and was conducted to a high ethical standard. Methods are described in detail and allow replication of the study by an independent researcher. However, I have three points here:

1. Coming back to the paper’s slight vagueness of how habitual motor responses, causal understanding, and conservatism interrelate, this vagueness seems to continue in the formulation of the research question. In the abstract, the authors state that they investigated if chimpanzees’ conservatism “could be attributed to the development of habitual motor responses when repeatedly solving a task”. This might suggest to the reader that the authors manipulated the formation of habitual motor responses in chimpanzees to see whether habit formation had an effect on conservatism. However, in the introduction (lines 90-95), the question is formulated differently, namely whether chimpanzees would form habitual motor patterns in the first place. This is a different (albeit not less interesting) research question. I urge the authors to change the respective part in the abstract and any other places in the manuscript to keep the formulation of this research questions consistent throughout. This is especially important because the validity of the task can only be assessed in the light of the question (i.e., if the aim had been to manipulate the formation of a habitual motor response, the experimental manipulation would have failed as the subjects showed a variety of strategies rather than habits).

2. The authors also mention that prior to their study, the formation of habitual motor responses had not been studied in chimpanzees nor gorillas. Given that knowledge about habit formation in chimpanzees is so limited, the authors could provide more information in the introduction as to why they have come to consider habitual motor patterns as a possible reason for chimpanzees’ conservatism. While some background information is given from lines 75-83, the introduction mainly focuses on the causal transparency/opacity aspect of a task; a couple of more sentences on what we (not yet) know about habit formation in chimpanzees would be useful, in order to be better able to locate the paper in the current literature and to clearly identify the current knowledge gaps. The authors could also consider moving the sentence from lines 102/103 to the previous paragraph.

3. A second research question is given in the introduction which is missing in the abstract, namely whether causal information can prevent conservatism. Again, this second question should be added to the abstract, as it seems to be a crucial question.
It does not become clear how the two research questions (or better: the concepts described in them) relate to each other. As suggested above, the authors should provide more information about the relationship between causal understanding, habits, and conservatism in the introduction.

Validity of the findings

The data have been provided. The data seem robust and all analyses to be statistically sound.

1a. I commend the authors for mentioning a possible limitation of their study, namely that the apes might not have gotten long enough exposure in order to form habits (lines 424-426). To follow up on this, might it not be possible that the Ape-plunk task is particularly unsuited for studying habit formation as it allows for such a variety of task solutions? Might the task actually be especially discouraging by design with regard to the formation of habits, and thus not only be less suited to study the research question as to whether apes form habits but also lack comparability to previously used tasks studying conservatism in apes? I suggest that the authors discuss the specific nature of the task they used.

1b. Somewhat related, the authors set out to investigate whether causal understanding of a task would prevent apes from becoming conservative; and while the data quite convincingly show that the authors succeeded in creating a task that was causally transparent for the apes, the Ape-plunk task also differs from previously used tasks in a second way, namely that it allows for a variety of strategies to be successful (24 in phase 1, 12 in phase 2). Causal transparency thus seems to be conflated with the number of strategies to be employed in phase 1. Therefore, can the authors really draw the conclusion that what prevented the apes from becoming conservative was the causal understanding of the task rather than the fact that they could use different solutions throughout the test and so were not “forced” by the task design to habitually use one or a more limited number of solutions? This question should also be addressed.

2. A point related to comparability of this study with previous ones: maybe a brief note should be made in the sentence from lines 427-428 (which states that in the Davis et al. (2016) study conservatism was found only after 20 trials) that the Davis et al. study did not allow for such a great variety of solutions.

3. Line 358-360: The authors suggest that the lack of habit formation might be explained by the apes’ causal understanding of the task. What about the possibility that there was a significant memory component involved, i.e., that apes simply forgot which specific strategy they had used in the previous trial? This memory demand becomes more pronounced when one considers the fact that the test was administered over several sessions. I would like to hear the authors’ opinion about this possibility.

Minor point: Discussion, line 322. The authors state “This suggests that there was no relationship between consistency of action sequences (habit formation) used in the first phase and ability to employ efficient action sequences when contingencies changed (flexibility).” I doubt whether the use of the term habit formation is warranted here as no habit formation was observed in phase 1. So wile it is valid to say that no relationship between consistency of action sequences in phase 1 and phase 2 was found, it is probably not valid to conclude that there is no relationship between habit formation and consistency of actions in phase 2 – as habit formation was not induced in this experiment.

Additional comments

This is a well-written, clear, and concise paper aiming to investigate the roles of causal understanding and habitual motor responses for chimpanzees’ conservatism in response to changing task demands. The authors contribute to the current debate by theorizing that chimpanzees’ conservatism found in several previous studies could result from the apes’ lack of causal understanding of the tasks. However, if the causal links between actions and the effects on the task are understood, the subject might be better able to evaluate the efficiencies of different actions and to switch to a more efficient strategy when task demands change. In order to investigate whether causal understanding of a task will prevent conservatism, the authors use a clever, new task, designed to be as causally transparent as possible. Indeed, the data suggest that the Ape-plunk task seems to be sufficiently causally transparent to the apes, with a large majority of the chimpanzees and gorillas solving the task efficiently from the first trial. The study also nicely demonstrates the apes’ flexibility in response to changing task demands. While there are no “uncorrectable issues” in the manuscript, some additional work should be done to elaborate on the suggested theory, especially how habit formation is thought to interplay with causal understanding and conservatism, as well as to clarify the research question regarding habit formation (i.e., is the question whether the apes form habitual motor responses or whether habitual motor responses affect flexibility?). Related to this, the authors should also discuss whether their task – whilst being a nice and clear problem-solving task which will likely be used in many future studies – is suited to address the question of the role of habit formation, as the variety of possible efficient solutions might heavily counteract the formation of habits. Some more work is also suggested on the discussion section, regarding the conclusions one can draw from the study. It remains unclear whether the apes’ lack of conservatism is due to the task’s causal transparency or the variety of possible solutions – these two aspects seem to be somewhat conflated in the task design. As the Ape-plunk task differs from previous studies not only in its causal transparency but also in the number of solutions possible, the authors should also discuss a bit more how their study relates to others.

·

Basic reporting

Literature references are sufficient.
Structure is correct.
Figures are relevant. Raw data are made available.
The work is self-contained with relevant results for hypotheses.

Experimental design

Research question is well defined. The gap in knowledge being investigated is clear.
Work was conducted rigorously up to a high technical and ethical standard.
Experimental design was well defined and sufficient for replication.

Validity of the findings

Data are robust, statistics are sound, but see below under general comments.
Conclusions are well stated and address the original research question. Conclusions are supported by results.

Additional comments

The manuscript is well written and reports some interesting results. Chimpanzees and gorillas quickly learned the task in phase 1 and used flexible responding when switched to the modified task in phase 2. The unexplainable top-down/bottom up species difference in phase 2 between chimpanzees and gorillas is intriguing. I have a few comments, mostly about the statistics

The authors do not state what the error bars represent in Figure 5. If they are SE, then it looks like there is a significant difference in Phase 1 because the chimpanzee and gorilla bars do not overlap. However, this rule of thumb only applies to a parametric test. Using a nonparametric test (Wilcoxon rank-sum test) the result did not reach statistical significance (p = 0.07; Lines 271-273). Since the authors used a nonparametric test, which compares medians, they should report the results using medians, not averages and SE, perhaps with boxplots. This is also true of any other tests in which averages are compared and reported using nonparametric tests.

If the authors really wish to report means (averages) as they do throughout results, they might check for assumption violations and use a parametric test (independent and paired t). If the authors are going to report averages they should also include standard deviations or SE each time (e.g., Line 272).

The authors state that the gorillas used the bottom up sequence (1, 2, 3, 4) in 26.6% of trials in phase one and that this was not significantly different than the chimpanzees, however they do not state the chimpanzees’ percentage (lines 281 – 284).

The authors report a p = 0.05 result as “did not differ” (Lines 326-327). Most hypothesis testing uses p ≤ 0.05 as the cutoff, so perhaps that should be reported as difference in run length between the phases?

Some of the gorillas seemed to be using a predominantly 1, 2, 3, 4 sequence in phase 1. If that was the case, the result might influence Discussion. The authors already state that the chimpanzees might have been using a simple associative rule “pull out the straw on which the reward is resting” to solve the phase 2 task without necessarily using cognitive flexibility (Lines 380 – 385). If a gorilla was using a bottom up strategy in phase 1, they would not need to make any cognitive flexibility adjustments moving from phase 1 to phase 2 to be successful. Perhaps this could be mentioned somewhere in Discussion.

Minor suggestions
Line 71: Add “cues” after causally-relevant.
Line 351: ‘a lot” is a bit colloquial; ‘a wide range of variation”?

---

## Round 0.2 · Minor Revisions

Thank you for being responsive to the comments during the last round of reviews. I think you have designed a very interesting task that can be applied to multiple questions and thus, would like to see it in the literature. I have only a few very minor suggestions before I can formally accept the MS.

Comments refer to the tracked changes version of your revised MS.
Is it possible that the chimpanzees and gorillas adopted different dominant strategies because of the opportunity to observe different strategies from their conspecifics?

On line 69, insert “was” before “prohibited.”

Watch for misplacement of the word “only.” Please move the only on line 81 to after “provide.” Please check carefully for other instances (line 93, 135 etc.).

On line 433, insert a comma after “level.”

Should there be an “as” before “what” on line 441.

It is nice that you thanked me and the reviewers for our comments; however, this is not necessary as it is our role to provide feedback.

---

## Round 0.3 · accepted · Accept

Thank you for fixing these last minor issues and thank you for a nice contribution to the literature!

#